

# Comparison of sagittal plane gait characteristics between the overground and treadmill approach for gait analysis in typically developing children

Rachel Senden[1], Rik Marcellis[1], Kenneth Meijer[2], Paul Willems[2], Ton Lenssen[1], Heleen Staal[3], Yvonne Janssen[4,5,6], Vincent Groen[3], Roland Jeroen Vermeulen[6] and Marianne Witlox[3]

[1] Department of Physical Therapy, Maastricht University Medical Center, Maastricht, Limburg, The Netherlands
[2] Department of Nutrition and Movement Sciences, NUTRIM School of Nutrition and Translational Research in Metabolism, Maastricht University Medical Center, Maastricht, Limburg, The Netherlands
[3] Department of Orthopaedic Surgery, Research School CAPHRI, Maastricht University Medical Center, Maastricht, Limburg, The Netherlands
[4] Centre of Expertise in Rehabilitation and Audiology, Adelante, Hoensbroek, Limburg, The Netherlands
[5] Department of Rehabilitation Medicine, School for Public Health and Primary Care, Maastricht University, Maastricht, Limburg, The Netherlands
[6] Department of Neurology, Maastricht University Medical Center, Maastricht, Limburg, The Netherlands

Corresponding author
Rachel Senden,
rachel.senden@mumc.nl

## ABSTRACT

**Background**. Instrumented treadmills have become more mainstream in clinical assessment of gait disorders in children, and are increasingly being applied as an alternative to overground gait analysis. Both approaches differ in multiple elements of set-up (*e.g.*, overground versus treadmill, Pug-in Gait versus Human Body Model-II), workflow (*e.g.*, limited amount of steps versus many successive steps) and post-processing of data (*e.g.*, different filter techniques). These individual elements have shown to affect gait. Since the approaches are used in parallel in clinical practice, insight into the compound effect of the multiple different elements on gait is essential. This study investigates whether the outcomes of two approaches for 3D gait analysis are interchangeable in typically developing children.

**Methods**. Spatiotemporal parameters, sagittal joint angles and moments, and ground reaction forces were measured in typically developing children aged 3–17 years using the overground (overground walking, conventional lab environment, Plug-In Gait) and treadmill (treadmill walking in virtual environment, Human Body Model-II) approach. Spatiotemporal and coefficient of variation parameters, and peak values in kinematics and kinetics of both approaches were compared using repeated measures tests. Kinematic and kinetic waveforms from both approaches were compared using statistical parametric mapping (SPM). Differences were quantified by mean differences and root mean square differences.

**Results**. Children walked slower, with lower stride and stance time and shorter and wider steps with the treadmill approach than with the overground approach. Mean differences ranged from 0.02 s for stride time to 3.3 cm for step width. The patterns of sagittal kinematic and kinetic waveforms were equivalent for both approaches, but significant differences were found in amplitude. Overall, the peak joint angles

were larger during the treadmill approach, showing mean differences ranging from 0.84° (pelvic tilt) to 6.42° (peak knee flexion during swing). Mean difference in peak moments ranged from 0.02 Nm/kg (peak knee extension moment) to 0.32 Nm/kg (peak hip extension moment), showing overall decreased joint moments with the treadmill approach. Normalised ground reaction forces showed mean differences ranging from 0.001 to 0.024.

**Conclusion**. The overground and treadmill approach to 3D gait analysis yield different sagittal gait characteristics. The systematic differences can be due to important changes in the neuromechanics of gait and to methodological choices used in both approaches, such as the biomechanical model or the walkway versus treadmill. The overview of small differences presented in this study is essential to correctly interpret the results and needs to be taken into account when data is interchanged between approaches. Together with the research/clinical question and the context of the child, the insight gained can be used to determine the best approach.

## INTRODUCTION

Gait abnormalities are known to occur in children with various pathological conditions or developmental disorders (*Lowenstein, Martin & Hauser, 2014*). Three-dimensional (3D) gait analysis is used for diagnostics and to evaluate interventions in these children (*Carollo, De & Akuthota, 2020*). Since milestones in the development of children are marked by changes in gait, normative gait data of typically developing (TD) children is essential for a correct interpretation of gait disorders. In addition, such normative data will help to better understand gait maturation during children's development.

Several approaches to 3D gait analysis are available. The overground approach is widely accepted and includes walking over a short walkway, frequently in combination with the Plug-in-Gait model (PiG) (*Baker et al., 2018*). Recently, instrumented treadmills embedded in a virtual environment and combined with dedicated kinematic models to compute real time kinematics, have become more mainstream (*Van den Bogert et al., 2013*; *Kainz et al., 2016*; *Flux et al., 2020*). Such a treadmill approach allows continuous recording of bilateral data for many steps. This increases data reliability (*Monaghan, Delahunt & Caulfield, 2007*) and creates the opportunity to study gait variability and stability (*McCrum et al., 2016*).

Approaches can differ in multiple elements of the set-up and workflow. Various kinematic models relying on different principles and assumptions can be used (*Flux et al., 2020*; *Falisse et al., 2018*). These models differ in *e.g.*, marker sets, hip joint center estimates, the use of optimization methods (global *vs.* local), preferred reference frame, whether the knee and ankle joints represent one degree of freedom or more, etc. Approaches can also differ in walking surface (*e.g.*, overground *versus* treadmill walking) (*Wearing, Reed & Urry, 2013*; *Jung et al., 2016*; *Gates et al., 2012*; *Kautz et al., 2011*; *Altman et al., 2012*; *Van der Krogt, Sloot & Harlaar, 2014*), walkway (*e.g.*, conventional lab *versus* virtual

environment) and consequently the number of strides recorded (*Sloot, Van der Krogt & Harlaar, 2014*). Studies have shown that these individual elements independently affect gait. The biomechanical model employed influences joint kinematics (*Flux et al., 2020*; *Falisse et al., 2018*) and kinetics (*Falisse et al., 2018*). *Flux et al. (2020)*, however, reported differences of 5° or less in sagittal plane kinematics across the Human Body Model (HBM), PiG and Calibrated anatomical system technique (CAST) model. Overground *versus* treadmill walking has been shown to involve small differences in spatiotemporal (*Wearing, Reed & Urry, 2013*; *Jung et al., 2016*; *Gates et al., 2012*; *Kautz et al., 2011*; *Altman et al., 2012*; *Van der Krogt, Sloot & Harlaar, 2014*), kinematic (*Jung et al., 2016*; *Van der Krogt, Sloot & Harlaar, 2014*) and kinetic parameters (*Riley et al., 2007*; *Watt et al., 2010*; *Parvataneni et al., 2009*). This was partly attributed to the fixed walking speed and enforced walking direction during treadmill walking. Minor adjustments in gait have been reported resulting from the presence of a virtual environment (*Sloot, Van der Krogt & Harlaar, 2014*). In addition, the post-processing of data can influence the outcome (*Rácz, 2021*). For instance, different filtering algorithms can be used , affecting step detection and the determination of specific events in the gait cycle. Furthermore, the innovative treadmill approaches, which include real-time feedback, have implemented several innovations in their computational methods in order to compute real-time kinematics and kinetics (*Van den Bogert et al., 2013*). For instance, technical markers and global optimization are implemented to minimize effects of marker dropout and soft tissue artefacts (*Duprey, Cheze & Dumas, 2010*).

In practice, approaches do not differ in just one element but simultaneously in multiple elements. Since different approaches are used in parallel in clinical and research settings, a comparison at the systems levels is warranted. It is important to examine possible differences in gait outcomes between a typical overground and an innovative treadmill approach. It is crucial to know whether outcomes are comparable or even interchangeable.

This study compared two established approaches to 3D gait analysis. It investigated whether the overground approach, in combination with the Plug-In Gait (PiG) model and a conventional lab environment, and the treadmill approach, in combination with the Human body model (HBM) and a virtual environment, result in differences in spatiotemporal parameters, sagittal joint angles and moments, and ground reaction forces (GRFs) in TD children aged 3–17 years. Meanwhile, a reference database of gait outcomes was created for TD children in both approaches.

## MATERIALS & METHODS

### Subjects

In this cross-sectional study with repeated-measures design, gait of 63 TD children was measured at the gait laboratories of the Maastricht University Medical Centre (MUMC+, Maastricht, The Netherlands). Inclusion criteria were: aged 3–17 years, able to walk independently, and absence of physical or mental impairments that interfere with walking ability. Six children were excluded because of technical problems ($n = 4$) and inability to complete the protocol ($n = 2$), leaving 57 children (25 boys/32 girls), with a mean age of

9.3 years (range 3–17 years). Kinetic data of one child was additionally excluded. Extended demographics are shown in Data S1.

All parents and children aged 12 years and older provided written informed consent prior to participation. The protocol was approved by the local Medical Ethics Committee of the MUMC+ (NL51929.068.14/METC142082).

## Procedure

A standardized physical examination according to *Becher et al. (2019)* was performed by an orthopedic resident to ensure that all included children had no physical impairments.

Data were collected as previously described at protocols.io (*Senden et al., 2020a*; *Senden et al., 2020b*). Specifically, first 3D gait analysis was performed using the overground approach (*Senden et al., 2020a*). Than the PiG marker set was expanded with additional markers on the jugular notch of the sternum, the xiphoid processus, the spinous processes cervical 7 and thoracic 10, the medial epicondyle of the femur, the medial malleoli, and the lateral side of the head of the 5th metatarsal, resulting in lower limb HBM-II with trunk markers (HBM) (*Flux et al., 2020*). Subsequently, 3D gait analysis was performed using the treadmill approach (*Senden et al., 2020b*). Only the comfortable walking speed is considered in this study, which was determined during at least five random overground walking trials by two movement detection portals placed 4 meter apart. Both measurements took approximately 1 hour.

## Data analysis

Data recording and processing is described at protocol.io (*Senden et al., 2020a*; *Senden et al., 2020b*). Data of sagittal joint angles, moments, and both vertical and anterior-posterior (AP) GRFs were extrapolated to strides (0–100%). Spatiotemporal parameters were determined for all valid strides. Subsequently, averages were calculated over the valid strides. Coefficient of variations of spatiotemporal parameters (standard deviation/mean * 100) were determined to assess gait variability (*Hausdorff, 2005*). For joint angles, moments and GRFs, the average over all valid strides was calculated for every percentage of the gait cycle. In addition, clinically relevant peak values in waveforms were determined (*Oudenhoven et al., 2019*). Only the stance phase was considered for joint moments and GRFs. GRFs were normalized for body weight according to *Hof (1996)*. Joint angles and moments were only evaluated in the sagittal plane. Within the gait research community, there is consensus that these are the most reliable (*Benedetti et al., 2017*).

The PiG and HBM model differ in several aspects. HBM uses multiple joint constraints. For instance, the HBM model constrains the knee and ankle joints to 1 and 2 degrees of freedom, respectively (*Flux et al., 2020*), thereby decreasing the effect of soft tissue artefacts and marker placement errors. PiG restricts segment movements to three degrees of freedom by using shared markers and joint centers between adjacent segments. PiG uses segment tracking, while HBM uses global optimization (*Duprey, Cheze & Dumas, 2010*), thereby further minimizing the effect of soft tissue artefacts and marker drop out. Furthermore, PiG defines hip joint center based on the Davis equation while the Harrington hip joint center equation is used in HBM (*Kainz et al., 2016*).

Spatiotemporal parameters, joint angles and moments and GRF of the right leg are presented for both approaches. The number of valid strides per child varied, so as many valid strides as possible were included.

## Statistics

Data is reported as mean and 95% confidence intervals (CI) and where appropriate in absolute numbers (*n*). Spatiotemporal and coefficient of variation (CoV) parameters, peak values in joint angles, moment and GRFs were compared using the repeated-measures paired t-tests or Wilcoxon signed-rank test, based on the normality of data tested with the Kolmogorov–Smirnov test. Mean differences (bias) and 95% CI of the difference were calculated (*Bland & Altman, 1986*). Joint angles, moments and GRFs waveforms of both approaches were compared using statistical parametric mapping (SPM), using a repeated-measures paired *t*-test. SPM analyses were implemented using the open-source spm1d code (v.M0.1, http://www.spm1d.org) in MATLAB (Mathworks, Natick, MA, USA). To quantify the differences in waveforms between the approaches, root mean square differences (RMSDs) were calculated over the whole gait cycle (*Flux et al., 2020*). RMSDs were calculated for each child and then averaged over all children. For kinematics, offset-corrected RMSDs (OC-RMSD) were calculated as the RMSD minus the offset, with offset defined as the difference in mean waveform (*Flux et al., 2020*). A difference in sagittal kinematics smaller than 5° was considered clinically irrelevant as this cut-off value corresponds to the measurement error for 3D gait kinematics (*Flux et al., 2020*; *Ferrari et al., 2008*; *Wilken et al., 2012*). To our knowledge, no such threshold has been defined for kinetics. All data was analyzed using the Statistical Package for the Social Sciences version 25 (SPSS, Chicago, Ill., USA). Significance level was set at $P \leq 0.05$.

## RESULTS

The calculation of spatiotemporal parameters for the overground and treadmill approaches was based on an average (range) of 42 (14–80) and 128 (98–224) strides, respectively, that for joint angles on 37 (14–68) and 106 (63–174) strides, that for joint moments on 7 (3–17) and 56 (5–141) strides and that for GRFs on 7 (3–17) and 55 (6–142) strides.

Significant differences in spatiotemporal parameters were found between the approaches. Although the comfortable speed determined during randomly selected overground trials was set as the treadmill speed, the actual speed was significantly lower during the treadmill approach (1.29 m/s (1.26–1.33 m/s) *vs.* 1.27 m/s (1.23–1.30 m/s)). Furthermore, a lower stride time, lower stance time but higher swing time, and shorter and wider steps were found during the treadmill approach. Mean differences ranged from 2.21 cm to 3.29 cm for spatial parameters and 0.02 s to 0.05 s for temporal parameters (Table 1). The coefficient of variation parameters were significantly larger for the overground approach (range CoV 4.65%–15.66%) compared to the treadmill approach (range CoV 0%–4.58%), except for CoV step width (Table 2).

Although the patterns of the waveforms of the sagittal joint angles (Fig. 1), GRFs (Fig. 2) and joint moments (Fig. 3) were similar with both approaches, significant differences in waveforms and peak values were found. Mean differences in joint angles (Table 1) and

**Table 1  Spatiotemporal parameters and peaks in sagittal joint angles and moments and GRFs for both approaches.**

| | Parameter | Overground approach Mean (95% CI) | Treadmill approach Mean (95% CI) | *P*-value | Mean difference (95%CI of difference) |
|---|---|---|---|---|---|
| Spatiotemporal | Walking speed (m/s) | 1.29 (1.26; 1.33) | 1.27 (1.23; 1.30) | 0.001* | 0.02 (0.01; 0.04) |
| | Stride time (s) | 0.93 (0.91; 0.96) | 0.91 (0.89; 0.94) | <0.001* | 0.02 (0.01; 0.03) |
| | Stance time (s) | 0.58 (0.56; 0.60) | 0.53 (0.51; 0.55) | <0.001* | 0.05 (0.04; 0.06) |
| | Swing time (s) | 0.35 (0.34; 0.36) | 0.39 (0.38; 0.39) | <0.001* | −0.03 (−0.04; 0.02) |
| | Step length (cm) | 60.37 (58.23; 62.50) | 58.16 (55.97; 60.35) | <0.001* | 2.21 (1.46; 2.96) |
| | Step width (cm) | 11.44 (10.60; 12.30) | 14.74 (13.63; 15.85) | <0.001* | −3.29 (−4.23; −2.35) |
| | Mean pelvic tilt | 9.59 (8.05; 11.13) | 10.43 (9.01; 11.85) | 0.099 | −0.84 (−1.84; 0.16) |
| Sagittal joint angles (°) | Peak hip extension terminal stance | 12.39 (10.51; 14.27) | 9.61 (7.96; 11.26) | <0.001* | 2.78 (1.28; 4.28) |
| | Peak hip flexion terminal swing | 33.32 (31.46; 35.19) | 38.82 (37.15; 40.48) | <0.001* | −5.49 (−7.16; −3.82) |
| | Knee flexion loading response | 20.95 (19.12; 22.77) | 26.15 (24.59; 27.70) | <0.001* | −5.20 (6.58; −3.82) |
| | Knee extension terminal stance | −4.37 (−2.97; −5.78) | −5.15 (−4.05; −6.26) | 0.234 | −0.78 (−0.52; 2.08) |
| | Peak knee flexion swing | 63.42 (61.86; 64.98) | 69.85 (68.88; 70.82) | <0.001* | −6.42 (−7.81; −5.03) |
| | Peak dorsal flexion terminal stance | 14.50 (13.43–15.56) | 10.71 (9.70–11.73) | <0.001* | 3.78 (2.74; 4.82) |
| | Peak plantar flexion terminal stance | 11.77 (10.39–13.14) | 15.35 (13.86–16.84) | <0.001* | −3.58 (−4.89; −2.27) |
| Normalised GRF (dimensionless) | Peak posterior force | 0.20 (0.19; 0.22) | 0.20 (0.19; 0.21) | 0.192 | 0.005 (−0.002; 0.012) |
| | Peak anterior force | −0.22 (−0.23; −0.20) | −0.21 (−0.22; −0.20) | 0.089 | −0.006 (−0.013; 0.001) |
| | Vertical peak force 1 | 1.19 (1.16; 1.23) | 1.19 (1.16; 1.22) | 0.530 | −0.001 (−0.001; −0.020) |
| | Vertical through force | 0.72 (0.69; 0.74) | 0.72 (0.70; 0.74) | 0.573 | −0.004 (−0.019; 0.010) |
| | Vertical peak force 2 | 1.10 (1.08; 1.13) | 1.08 (1.06; 1.10) | 0.001* | 0.024 (0.010; 0.038) |
| Sagittal moments (Nm/kg) | Peak hip extension moment | 0.97 (0.89; 1.05) | 0.65 (0.60; 0.69) | <0.001* | 0.32 (0.24; 0.40) |
| | Peak hip flexion moment | −0.95 (−1.00; −0.89) | −0.74 (−0.78; −0.69) | <0.001* | −0.21 (−0.25; −0.17) |
| | Peak knee extension moment | 0.53 (0.43; 0.62) | 0.51 (0.47; 0.56) | 0.735 | 0.02 (−0.08; 0.11) |
| | Peak ankle plantar flexion moment | 1.27 (1.21; 1.33) | 1.31 (1.24; 1.38) | 0.017 | −0.04 (−0.07; −0.01) |

**Notes.**
*Sign. difference (*P* < 0.05) between the approaches.
CI, confidence interval.

**Table 2** Coefficient of variation for spatiotemporal parameters for both approaches.

| (%) | Overground approach | | Treadmill approach | | *P*-value |
|---|---|---|---|---|---|
| | Mean | 95% CI | Mean | 95% CI | |
| CoV speed | 15.66 | 14.62–16.71 | 0.00 | 0.00–0.00 | <0.001 |
| CoV stride time | 4.65 | 3.89–5.40 | 3.07 | 2.72–3.42 | <0.001 |
| CoV stance time | 7.45 | 6.48–8.43 | 4.41 | 3.95–4.86 | <0.001 |
| CoV swing time | 9.17 | 5.62–12.73 | 2.93 | 2.67–3.18 | <0.001 |
| CoV step length | 7.78 | 7.24–8.33 | 4.58 | 4.10–5.06 | <0.001 |
| CoV step width | 28.19 | 25.59–30.78 | 27.42 | 24.39–30.46 | 0.597 |

Notes.
*Sign. difference ($P < 0.05$) between the approaches.
CI, confidence interval.

**Table 3** Root mean square differences in sagittal plane kinematic, kinetic and GRF waveforms between the approaches, calculated over the averaged gait cycle.

| | Joint | RMSD Mean (95%CI) | OC-RMSD Mean (95%CI) | Offset Mean (95%CI) |
|---|---|---|---|---|
| Sag. Joint angles (°) | Pelvis | 2.98 (2.30; 3.66) | 0.17 (0.13; 0.21) | 2.81 (2.11; 3.51) |
| | Hip | 7.91 (6.76; 9.07) | 2.68 (2.30; 3.06) | 5.23 (3.88; 6.59) |
| | Knee | 9.04 (8.12; 9.95) | 4.39 (3.89; 4.90) | 4.64 (3.53; 5.75) |
| | Ankle | 5.45 (4.92; 6.05) | 1.98 (1.71; 2.25) | 3.48 (2.79; 4.18) |
| GRFs (Dimensionless) | AP | 0.022 (0.020; 0.025) | | |
| | Vertical | 0.080 (0.074; 0.086) | | |
| Sag. Joint moments (Nm/kg) | Hip | 0.25 (0.22; 0.27) | | |
| | Knee | 0.18 (0.15; 0.21) | | |
| | Ankle | 0.12 (0.10; 0.13) | | |

Notes.
RMSD, root mean square difference; OC, offset-corrected; CI, confidence interval; GRF, ground reaction forces; AP, anterior-posterior; Vert, vertical; Sag, sagittal.

their offset-corrected root mean square differences (OC-RMSDs, Table 3) were less than 5°. Except the peak knee flexion during loading response and swing phase and the peak hip flexion during terminal swing which showed mean differences of 5.20°, 6.42° and 5.49°, respectively. Significant higher normalized second vertical peak force was found during the overground approach, showing a mean difference of 0.024 (Table 1). The peak hip extension and flexion moment during stance was significantly higher with the overground approach showing mean differences of 0.32 Nm/kg and 0.21 Nm/kg, respectively (Table 1). However, overall hip extension moment was significantly lower during the whole stride with the overground approach (Fig. 3). SPM showed significantly higher knee flexion moment during terminal stance for the overground approach (Fig. 3), while peak knee moments were similar (0.53 Nm/kg for the overground approach and 0.51 Nm/kg for the treadmill approach). Peak ankle plantar moment during terminal stance was significantly lower during the overground approach, showing mean difference of 0.04 Nm/kg (Table 1).

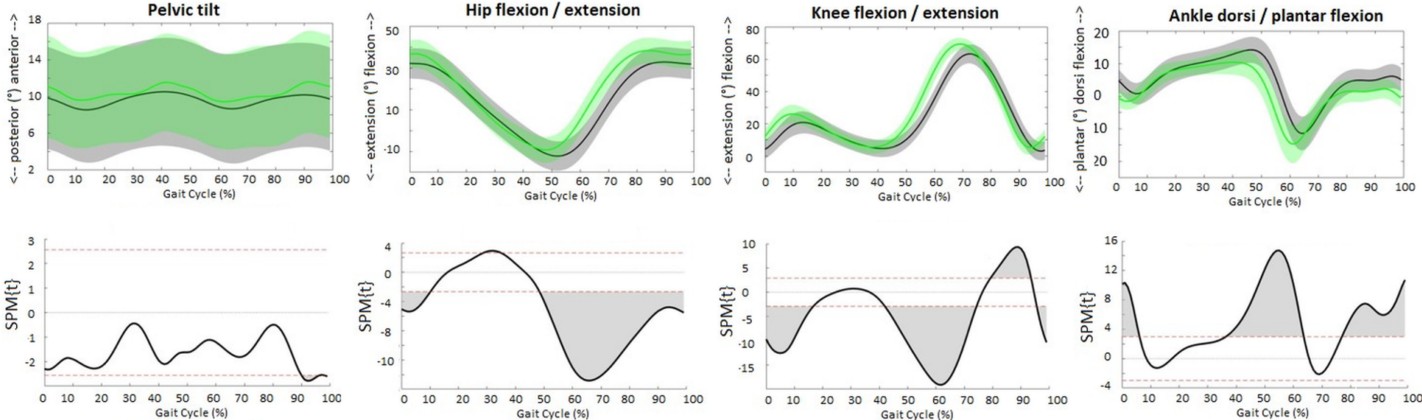

**Figure 1 Sagittal joint angles.** The upper figures present the averaged waveforms and standard deviations of pelvis, hip, knee and ankle angle in sagittal plane over all subjects for the overground (grey) and treadmill (green) approach. The lower figures present the SPM paired $t$-test analyses. Red dotted lines indicate $t$-threshold values above/below which curves significantly differ. Gray shaded areas indicate significant differences between the two approaches.

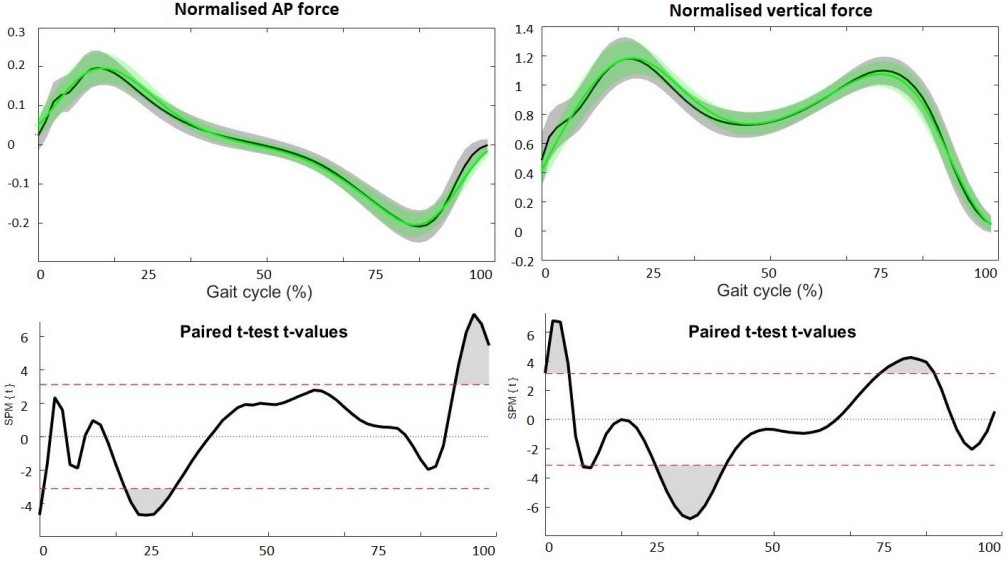

**Figure 2 Ground reaction forces.** The upper figures present the averaged waveforms and standard deviations of AP (left) and vertical (right) GRF during stance phase for the overground (grey) and treadmill (green) approach. The lower figures present the SPM paired $t$-test analyses. Red dotted lines indicate $t$-threshold values above/below which curves significantly differ. Gray shaded areas indicate significant differences between the two approaches.

## DISCUSSION

This study compared the outcomes of the overground and treadmill approach, two widely accepted but methodologically different approaches to 3D gait analysis in TD children aged 3 to 17 years. Although both approaches are suitable for 3D gait analysis in TD children,

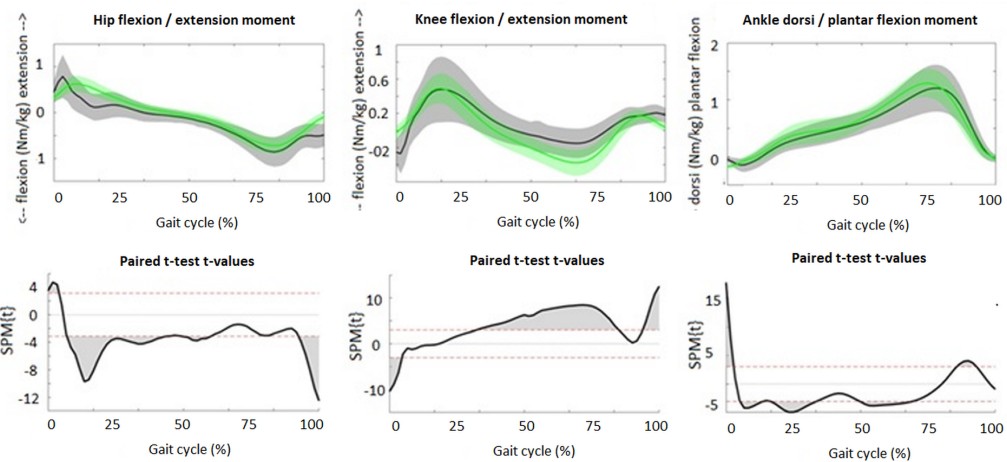

**Figure 3  Sagittal plane joint moments.** The upper figures present the averaged waveforms and standard deviations of hip, knee and ankle moments in sagittal plane during stance phase for the overground (grey) and treadmill (green) approach. The lower figures present the SPM paired $t$-test analyses. Red dotted lines indicate $t$-threshold values above/below which curves significantly differ. Gray shaded areas indicate significant differences between the two approaches.

significant differences were found in spatiotemporal and coeffient of variation parameters, sagittal joint angles and moments and GRFs. This makes that data is not interchangeable between approaches. The overview of small differences can be used clinically to select an approach and to critically interpret the results.

The observed discrepancies were in line with those reported in studies investigating the influence on gait of one single element. As regards spatiotemporal parameters, the 4% decrease in step length and the 3.3 cm increase in step width found in the treadmill approach corresponds to the results of previous studies, which reported 3% and 7% reductions in stride length in TD children (*Van der Krogt, Sloot & Harlaar, 2014*) and healthy elderly persons, respectively (*Watt et al., 2010*), and a 2.9 cm increase in step width in TD children (*Van der Krogt, Sloot & Harlaar, 2014*) when walking on a treadmill. Regarding sagittal joint angles, nearly identical waveform patterns were found, although there were significant differences in amplitude. The peak hip extension was approximately 3° less with the treadmill approach, which agrees with the 3° and 4° reductions reported by *Watt et al. (2010)* and *Van der Krogt, Sloot & Harlaar (2014)* in healthy elderly persons and TD children, respectively, when comparing overground with treadmill walking. *Flux et al. (2020)* found differences smaller than 5° in sagittal hip and knee angles across models, but differences up to 7.3° in peak ankle dorsiflexion. We found the largest differences in peak knee and hip flexion during swing phase, which were 6.4° and 5.5° greater, respectively, with the treadmill approach. This is comparable to the findings of Jung et al., who reported an increase of 5.1° and 4.6°, respectively, during treadmill walking by children with cerebral palsy (CP) (*Jung et al., 2016*). However, they contrast with those of Riley et al., who reported smaller hip flexion (mean difference 0.64, $p = 0.02$) and knee flexion (mean difference 0.68, $p = 0.06$) for treadmill walking (*Riley et al., 2007*). Although
the differences in joint angles we found were admittedly significant, most differences, except the peak knee flexion during loading response and swing phase and the peak hip flexion during swing phase, fell within the normal variability of gait, *i.e.,* less than those associated with a 5° measurement error (*Wilken et al., 2012*). *Wilken et al. (2012)* reported minimal detectable changes for intra- and interrater-intersession comparisons of 5.80° and 7.33° for peak hip and knee flexion during swing respectively. This can explain why these parameters showed the largest difference between the two approaches. As regards GRF and sagittal joint moments, significantly lower amplitudes were found for the treadmill approach. We found a 2% lower second peak in vertical GFR in the treadmill approach than the overground approach. This is in line to previous studies reporting GRFs as a function of body weight, showing lower amplitudes in second peak vertical GRF in treadmill walking compared to overground walking in healthy adults, with mean differences ranging from 5.5–7.8 (*Riley et al., 2007*; *Parvataneni et al., 2009*). Those studies also reported lower peak posterior GRF for the treadmill approach (mean difference range 1.4–4.4), which contrasts our findings which show similar peak posterior GRF for both approaches. The 3% smaller plantar flexion moment for the overground approach and the observed mean difference of 0.21 Nm/kg in hip flexion moment between the two approaches is in line with the study of *Lee & Hidler (2008)* who compared overground with treadmill walking in healthy adults, showing a 2% smaller plantar flexion moment for the treadmill walking and a mean difference of 0.13 Nm/kg in hip flexion moment. The differences in joint moments we found did not exceed the minimal detectable change reported by *Wilken et al. (2012)* (range 0.00–0.31 Nm/kg), who interpreted them as being not sufficiently different to be relevant.

In addition to the findings of previous studies which have already demonstrated an effect on gait of individual elements (*e.g.*, treadmill *vs.* overground, the biomechanical model), the present study shows that the compound effect on gait of the multiple different elements falls within the normal variability of gait (*Riley et al., 2007*; *Watt et al., 2010*). The systematic differences that are observed can be attributed to important changes in the neuromechanics of gait. For instance, the increased step width may be adapted to imply changes in stability control. In addition the differences can be due to methodological choices used in both approaches such as the set-up (*e.g.*, overground *vs.* treadmill or biomechanical model), workflow (*e.g.*, the amount steps recorded and included in the analysis) and post-processing of data (*e.g.*, filtering characteristics), as described earlier (*Wearing, Reed & Urry, 2013*; *Jung et al., 2016*; *Gates et al., 2012*; *Kautz et al., 2011*; *Altman et al., 2012*; *Van der Krogt, Sloot & Harlaar, 2014*; *Sloot, Van der Krogt & Harlaar, 2014*; *Riley et al., 2007*; *Watt et al., 2010*; *Parvataneni et al., 2009*). Treadmill walking has the benefit of recording many successive steps. This reduces the variability of gait which is indicated by the smaller coefficients of variation for the treadmill approach. The impact of a misstep is than unneglectable, in contrast to overground walking where one misstep has a large impact due to the limited amount of strides recorded. *Monaghan, Delahunt & Caulfield (2007)* showed that data reliability increases when multiple steps are collected. The ability to acquire larger data sets is therefore an inherent advantage of the treadmill approach (*Riley et al., 2007*).

Although both approaches are suitable for 3D gait analysis in TD children, one should account for the small differences when interchanging gait characteristics of both approaches. This information is essential, since both approaches are used in parallel in everyday practice. In view of the small systematic differences that are reported between the two approaches, we recommend collecting normative data for both approaches separately. In addition, the overview of small differences reported in this study can be useful as a reference for choosing and interpreting data of one of these two approaches. The choice of one approach should be based on the research or clinical question that needs to be answered and the context of the child. For instance, if gait variability or fatigue is the aspect of interest, gait should be measured at the treadmill, as this allows the recording of many successive steps. If the question concerns hampered gait in severely affected children with CP, treadmill walking might be too stressful, more cognitively demanding and technically infeasible, making the overground approach more suitable.

Several limitations of this study need to be considered. First, the waveforms produced can be subject to shifts in time as a result of event detection and filtering characteristics. On average, the peaks in waveforms of the treadmill approach occur earlier than those of the overground approach, showing an average phase shift of 3% (range 0–5%). Additional analysis, where SPM analyses were performed with phase-shift corrected waveforms, showed that this phase shift had no impact on the findings (see Data S1). Second, although the walking speed was intended to be identical for both approaches, it was slightly but significantly faster during the overground approach. This can be explained by the fact that the comfortable walking speed was determined during random overground walking trials, which may be different from the trials selected for the analyses. Although higher speed may have an effect on gait (*Fukuchi, Fukuchi & Duarte, 2019*), the influence of the higher speed during overground walking was presumably minimal, as the difference in speed was very small (<0.02 m/s). Third, the analysis of kinetics relied on a small number of strides in a few children, which may cause type II errors. Specifically, the lowest amount of strides used for analyses was three, which was observed in one child for the calculation of kinetics for the overground approach (*Flux et al., 2020*). At least five strides were used to calculate kinetics in 74% (overground approach) and 98% (treadmill approach) of the children. Spatiotemporal and kinematic variables were based on at least 20 strides in respectively 97% and 95% of the children for the overground approach and in all children for the treadmill approach. The amount analyses relying on a low number of strides is thus limited. Therefore the chance for type II errors is minimal. Fourth, only the sagittal plane was evaluated. Frontal and transverse plane joint angles and moments need to be considered in future studies. Fifth, there may have been an effect of fatigue. Both measurements took approximately one hour, and the treadmill approach was always performed last. This sequence was chosen for practical reasons, and was the same for all children. Finally, we mainly considered group differences. A more in-depth analyses at individual level is recommended.

## CONCLUSIONS

Although the overground and treadmill approaches are both suitable for 3D gait analysis in TD children, they produce different sagittal gait characteristics. It is essential to account for these differences when data is interchanged between the two approaches. The overview of the small differences between approaches presented here, together with the research/clinical question and the context of the particular child, can be used to select the best approach and to critically interpret the results.

## ACKNOWLEDGEMENTS

The authors like to thank K Theunissen, M Coenen, W Bijnens, I Moll, C McCrum and T Smeijsters, for helping to gather and process the data.

### Funding
The authors received no funding for this work.

### Competing Interests
The authors declare there are no competing interests.

### Author Contributions
- Rachel Senden conceived and designed the experiments, performed the experiments, analyzed the data, prepared figures and/or tables, authored or reviewed drafts of the article, and approved the final draft.
- Rik Marcellis conceived and designed the experiments, performed the experiments, authored or reviewed drafts of the article, and approved the final draft.
- Kenneth Meijer conceived and designed the experiments, authored or reviewed drafts of the article, and approved the final draft.
- Paul Willems analyzed the data, authored or reviewed drafts of the article, and approved the final draft.
- Ton Lenssen authored or reviewed drafts of the article, and approved the final draft.
- Heleen Staal conceived and designed the experiments, authored or reviewed drafts of the article, and approved the final draft.
- Yvonne Janssen performed the experiments, authored or reviewed drafts of the article, and approved the final draft.
- Vincent Groen performed the experiments, authored or reviewed drafts of the article, and approved the final draft.
- Roland Jeroen Vermeulen conceived and designed the experiments, authored or reviewed drafts of the article, and approved the final draft.
- Marianne Witlox conceived and designed the experiments, authored or reviewed drafts of the article, and approved the final draft.

## Human Ethics

The following information was supplied relating to ethical approvals (*i.e.*, approving body and any reference numbers):

The study was approved by the local Medical Ethics Committee of the MUMC+ (NL51929.068.14/METC142082).

## Data Availability

The raw measurements are available in the Supplementary Files.

## Supplemental Information

Supplemental information for this article can be found online at http://dx.doi.org/10.7717/peerj.13752#supplemental-information.

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
