# Peer review of "Comparison of sagittal plane gait characteristics between the overground and treadmill approach for gait analysis in typically developing children"

_PeerJ, doi:10.7717/peerj.13752_

## Round 0.1 · original submission · Major Revisions

Reviewer 2 noted that he was concerned about the low number of strides used for analysis. Please make sure to address this. My additional concern would be that some large and clinically relevant differences may have been not significant, due to the small number of cycles. Please try to discuss the Type 2 error rate for the study (in Discussion).

Reviewer 2 also suggested to look at non-sagittal variables. I think this would be good and useful, but you may decide that this is outside of the scope of the study, or mention that this still needs to be done.

And also a small technical correction, you should say that "HBM model constrains the knee and ankle joints to 1 and 2 degrees of freedom, respectively". Indeed this makes it difficult to compare non-sagittal variables. But you could definitely do so for the hip angles and moments, and for the knee moments, which are available in HBM as a 3D intersegmental moment. If you decide to compare 3D moments, watch out for different reporting standards being used between the two software systems (Derrick et al., J Biomech 2020).

Reviewer 1 ·

Basic reporting

The reporting is sufficient although the use of subjective words like 'small' should be avoided.

Experimental design

The experimental design is well-justified and the methods are described in sufficient detail. Suggest adding an analysis of the difference in variation between the two modalities.

Validity of the findings

The findings are valid and well described. However, the interpretation is problematic. Please see additional comments for more information.

Additional comments

This study used a repeated-measures design to examine differences between overground and treadmill walking. The authors examined differences at the system level – different underlying models were also used in addition to walking overground vs. a treadmill.
The rationale for looking at system-wide differences rather than changing one parameter is well justified and the analyses of both peak values and waveforms using SPM are appropriate. My primary concern is sending a mixed signal to the readers regarding whether the systems can be interchangeable. On the one hand, the authors say that the differences are small and clinically insignificant. While the authors were careful on how clinical significance was defined, a 5-degree difference may be important in certain circumstances, especially given that this is a repeated measures design. The field is not far enough to make a context-independent claim on clinical significance even if this is within measurement error. On the other hand, the authors state that there are systematic differences and recommend generating different databases of typically developing children. While it is not possible to tease out whether the observed differences result from methodological choice or changes in the neuromechanics of gait with this paradigm, it is suggested that the authors emphasize the differences and let readers decide whether they can interchange modalities for their application and research question. I am concerned that, as written, this manuscript will allow researchers to interchange the results from modalities using this manuscript citation as the rationale. In reality, there are important differences suggestive of important changes in neuromechanics (e.g. increases in step width imply changes in stability control). Additionally, the authors did not examine out-of-plane measurements, and this should be emphasized in the title.
Line by Line Comments
Title: Add ‘sagittal’
Abstract
Line 32-33: This sentence is unclear.
Line 34: ‘…two methodological approaches…’ Just state the approaches.
Line 35: suggest avoiding subjective phrases like ‘comparable’.
Abstract – Methods
Using SPM is a strength of this study. Suggest including it in the methods of the abstract.
Line 43-45: Suggest replacing vague and/or subjective words like ‘differ’, ‘small’, and ‘very similar’ with actual values and let the reader decide.
Line 46-47: Is there a way to specify what is meant by overground and treadmill approach given that is general and the authors tested a very specific instantiation of both modalities?
Introduction
Line 59: ‘…, mostly in combination with the Plug-in-Gait model,…’ Please check the accuracy of this statement. To my knowledge, there are many labs that use different models (e.g. 6dof cluster tracking) for clinical gait analysis.
Line 60: Plug in gait is not a gold standard.
Line 69: ‘…as regards..’ is somewhat awkward phrasing.
Line 72-75: Please delete ‘only’ from ‘only 5 degrees’. Depending on the research question, this can be a large change in motion.
Line 77: ‘…various ways’ is vague.
Line 80: Please check grammar.
Line 82-83: ‘certain mechanical effects’ and ‘affect gait outcomes’ is vague. Please be more specific.
Methods
Suggest stating the number of subjects in the subjects section of the methods. Repeated measures is a strength of the study, suggest emphasizing it in the methods. Does using different numbers of cycles between modalities affect the statistics. Suggest using the lowest number of cycles for everything. It looks like the variation is different between the modalities. Suggest quantifying and reporting this.
Results
Suggest including the range of ages instead of the CI.
Please avoid words like ‘small’.
Lines 173: Please write out ‘OC-RMSD’. Avoid the word ‘only’. If the study set out trying to show an effect, this would be phrased differently.
Lines 191-192: The knowledge is not interchangeable.
Line 213: As mentioned, clinically irrelevant is too strong a claim.
Line 217: Delete ‘highly similar’.
Line 230: Mentioning the differences are within the normal variability of gait is important. Suggest moving this up earlier and using it as an outcome measure. That said, it is a repeated-measures study so this interpretation is somewhat limited.
Line 246: Duplicate sentences.
Line 248-249: Please provide more details on what the additional analysis was.
Line 255: change ‘transversal’ to ‘transverse’.
Conclusions: Please use more careful wording. It also does not make sense to first say you can use them interchangeably and then recommend separate normal databases. Suggest emphasizing the latter.

·

Basic reporting

The article is clear and reads very well. Figures are good.

The authors might consider using % of the stance phase (0-100) as x-axis in their figures, instead of % of the gait cycle (0-60), but I leave it up to their appreciation.

Experimental design

The authors provide little insights into why small differences exist between both approaches. They acknowledge their existence, report about their clinical implication, and suggest contributing factors, but they do not probe further. Although I am not suggesting that they should try to quantify the effect of each single element, some sensitivity analyses seem accessible (eg, processing overground and treadmill data with the same system, use more similar biomechanical models, etc.) and would nicely complement those results.

Since gait abnormalities can manifest in other planes than the sagittal plane. It would be valuable to consider studying kinematics and kinetics in other planes too. The authors mention that sagittal plane kinematics and kinetics are more reliable, but it would still be valuable to know whether the two approaches differ for out of the sagittal plane degrees of freedom. If they do, this would suggest users should be cautious about over-interpretation of those results. This would be a valuable contribution. The authors mention that studying the sagittal plane only is a limitation of the study, but it seems like a low-hanging fruit that they should consider going for as they should have the data already.

Validity of the findings

I am deeply confused about the averages and ranges provided on lines 159-162. Why aren’t all strides used for calculating the different variables? Why are the numbers varying so much in between variables (eg, GRF vs kinematics vs kinetics)? In some cases, only 3 strides are used (for some joint moments). That seems very few to bring confidence in the results, and not enough to enable statistical analysis.

Additional comments

Minor comments:

Line 72: Since Flux et al. likely did not compare across all models, please consider making this statement less general. This is only one available comparison.
Line 100: Please provide some quantitative information about the studied population (number of subjects, demographics, etc) in the methods rather than in the results.
Line 130-131: Can you specify how/if that is different from the plug-in-gait model (for non-experts)? Differences in joint definitions might lead to different outcomes. The authors should consider adding differences in biomechanical models in their list lines 230-233.
Line 133: Why did you include both legs for the GRFs and not for the kinematics/kinetics?
Line 246: twice the same sentence.
Line 267: missing bit of sentence.

---

## Round 0.2 · Minor Revisions

Please submit another revision to address the remaining comments of reviewer 2. I will probably make the final decision without sending it back to the reviewer.

·

Basic reporting

Thanks for answering my questions. I have one more.

The authors support some of their methodological choices by citing other papers, which is of course a good practice. However, it does not sound right in certain cases. For instance, they justify that their number of strides is large enough because someone did a study with fewer strides. This is not a good justification. It is not because someone did it that it is good to do it. I would encourage the authors to review those statements. For example: "Specifically, the lowest amount of strides used for analyses was three, which was observed in one child for the calculation of joint moments. Three gait cycles were also used in the study of Flux et. al reporting on 3D gait analysis using different biomechanical models (6)."

Experimental design

Thanks for answering my questions. I have one more.

I previously asked: "Line 133: Why did you include both legs for the GRFs and not for the kinematics/kinetics?"
And you answered: " To increase the number of valid strides and subsequently consistency, we included the left and right leg for GRF data. Since no differences between left and right leg were found (supplementary data), we only included data of the right leg for spatiotemporal, joint angles and joint moments."

This does not seem very consistent to me. Since you found no difference between both legs, you should consider only including right leg data for all measures of interest. As it is, it is not very rigorous. This is minimal change as it only concerns the GRF.

Validity of the findings

No further questions. Thanks.

Additional comments

Thanks for answering my questions. I would have liked to see some of my suggestions implemented (eg, reporting on frontal plane measures, performing some sensitivity analysis), since it would have only required further data processing and no additional data collection and generation. However, I understand if you consider it is out of the scope of the paper. Thank you for addressing my comments.

---

## Round 0.3 · accepted · Accept

Thank for addressing the reviewer comments in your revision. The manuscript is now accepted for publication.